# Genome-Wide Identification and Involvement in Response to Biotic and Abiotic Stresses of lncRNAs in Turbot (*Scophthalmus maximus*)

**DOI:** 10.3390/ijms242115870

**Published:** 2023-11-01

**Authors:** Weiwei Zheng, Yadong Chen, Yaning Wang, Songlin Chen, Xi-wen Xu

**Affiliations:** 1State Key Laboratory of Mariculture Biobreeding and Sustainable Goods, Yellow Sea Fisheries Research Institute, Chinese Academy of Fishery Sciences, Qingdao 266071, China; ydzhengweiwei@163.com (W.Z.); chenyd@ysfri.ac.cn (Y.C.); wwwwynyy66@163.com (Y.W.); 2Laboratory for Marine Fisheries Science and Food Production Processes, Laoshan Laboratory, Qingdao 266237, China; 3College of Life Science, Qingdao University, Qingdao 266071, China

**Keywords:** *Scophthalmus maximus*, long non-coding RNAs (lncRNAs), abiotic stress, biotic stress

## Abstract

Long non-coding RNAs (lncRNAs) play crucial roles in a variety of biological processes, including stress response. However, the number, characteristics and stress-related expression of lncRNAs in turbot are still largely unknown. In this study, a total of 12,999 lncRNAs were identified at the genome-wide level of turbot for the first time using 24 RNA-seq datasets. Sequence characteristic analyses of transcripts showed that lncRNA transcripts were shorter in average length, lower in average GC content and in average expression level as compared to the coding genes. Expression pattern analyses of lncRNAs in 12 distinct tissues showed that lncRNAs, especially lincRNA, exhibited stronger tissue-specific expression than coding genes. Moreover, 612, 1351, 1060, 875, 420 and 1689 differentially expressed (DE) lncRNAs under *Vibrio anguillarum*, *Enteromyxum scophthalmi*, and Megalocytivirus infection and heat, oxygen, and salinity stress conditions were identified, respectively. Among them, 151 and 62 lncRNAs showed differential expression under various abiotic and biotic stresses, respectively, and 11 lncRNAs differentially expressed under both abiotic and biotic stresses were selected as comprehensive stress-responsive lncRNA candidates. Furthermore, expression pattern analysis and qPCR validation both verified the comprehensive stress-responsive functions of these 11 lncRNAs. In addition, 497 significantly co-expressed target genes (correlation coefficient (*R*) > 0.7 and *q*-value < 0.05) for these 11 comprehensive stress-responsive lncRNA candidates were identified. Finally, GO and KEGG enrichment analyses indicated that these target genes were enriched mainly in molecular function, such as cytokine activity and active transmembrane transporter activity, in biological processes, such as response to stimulus and immune response, and in pathways, such as protein families: signaling and cellular processes, transporters and metabolism. These findings not only provide valuable reference resources for further research on the molecular basis and function of lncRNAs in turbot but also help to accelerate the progress of molecularly selective breeding of stress-resistant turbot strains or varieties.

## 1. Introduction

Long non-coding RNAs (lncRNAs) are a kind of RNA molecule whose transcript is greater than 200 bp but without protein-coding potential [1]. However, lncRNAs have many similar features to mRNA, such as 3′ polyadenylation, 5′ capping structure and RNA splicing, but have few or no open reading frames [2,3]. In addition, lncRNAs usually show lower expression levels and sequence conservation, but they exhibit stronger cell-specific and tissue-specific expression patterns compared to mRNAs [4,5]. In general, most lncRNAs are transcribed through RNA polymerase II and mature after splicing [6,7]. According to their positions and directions of transcription relative to protein-coding genes, lncRNAs are generally divided into intronic lncRNAs, antisense lncRNAs, intergenic lncRNAs (lincRNAs), and overlapping lncRNAs [8,9].

With the development of bioinformatics, whole genome sequencing technology, and a great deal of available RNA-seq datasets resources, genome-wide identification of lncRNAs has been conducted in plenty of species including *Arabidopsis thaliana* [10,11], *Drosophila* [12], rat [13], and human [14]. Moreover, lncRNAs have also been systematically identified in many fish species at the genome-wide level, such as *Oncorhynchus mykiss* (rainbow trout) [15,16], *Genypterus chilensis* (red cusk-eel) [17], *Salmo salar* (Atlantic salmon) [18], *Oreochromis niloticus* (tilapia) [19], *Danio rerio* (zebrafish) [20], *Larimichthys crocea* (large yellow croaker) [21], *Oncorhynkus kisutch* (Coho salmon) [22,23], *Tetraodon nigroviridis* (green spotted puffer) [24], and *Cyprinus carpio* (common carp) [25]. However, genome-wide identification of lncRNAs in turbot has not been reported to date.

LncRNAs were once considered to be junk RNA or background transcriptional noise with no biological functions [26]. However, an increasing number of studies have recently shown that lncRNAs play crucial regulatory roles in various biological processes, such as transcriptional regulation [27], post-transcriptional gene regulation [28], epigenetic regulation [29], proliferation [30], senescence [31,32], immune responses [33], quiescence, growth [30,34] and stress response [35]. Notably, some recent studies have also proven the potential roles of lncRNAs in response to various abiotic and biotic stresses in a variety of fish species. For instance, a total of 428 DE lncRNAs were identified in the liver of *O. mykiss* under heat stress, and the lncRNA-mRNA regulatory network further provided insights into the regulatory roles of lncRNAs on gene expression in *O. mykiss* under heat treatment stress [15]. Moreover, 112, 323, and 108 DE lncRNAs were identified in the skeletal muscle, head kidney, and liver tissues of *G. chilensis*, respectively, in response to handling stress. The co-expression network analysis provided valuable information regarding the relationship between handling stress and lncRNAs [17]. Li et al. identified 99 DE lncRNAs in response to hypoxia, salt, and cold stress, which laid an important foundation for the elucidation of the molecular regulatory mechanism of lncRNAs in abiotic stress response in tilapias [19]. Wang et al. identified 226 DE lncRNAs in the skeletal muscles of juvenile rainbow trout exposed to estradiol and further demonstrated the molecular regulatory mechanism of lncRNAs in response to estradiol exposure stress through the lncRNA-mRNA co-expression network [36]. A total of 163 lncRNAs specifically expressed in the spleen of *L. crocea* under *Vibrio parahaemolyticus* infection stress were identified, which indicated their involvement in the immune response in *L. crocea* [37]. Differentially expressed lncRNAs were detected in Atlantic salmon under salmon anemia virus (ISAV) infection stress, indicating that these lncRNAs may be involved in the regulation of host responses to ISAV infection stress [38].

Turbot (*Scophthalmus maximus*), an economically important cold-water flatfish species, has become a worldwide marine-culture fish species with high nutritional value and delicious taste [39]. Unfortunately, because of high density intensive culture and the frequent occurrence of extreme weather events, the turbot has been exposed to various abiotic and biotic stresses in the process of breeding, such as heat [40], salinity [41,42], oxygen [43], and multiple pathogen infection stresses [44,45,46,47]. These environmental stresses severely threatened the healthy and survival of turbot, which has led to enormous economic losses to the turbot aquaculture industry and greatly hindered the health and sustainable development of the turbot aquaculture industry. A large number of studies have been conducted on the changes in gene expression under various environmental stresses in recent years [48,49,50]. However, there has been no systematic identification and characterization of lncRNAs in turbot and no adequate detection of their roles in response to abiotic and biotic stress in turbot up to now. Fortunately, abundant available RNA-seq dataset resources [40,41,42,43,45,46,47,51,52] and high-quality turbot genome sequences [50] make it possible to conduct systematic identification, characterization, and functional study of lncRNAs in the turbot.

In this study, to identify a more complete lncRNAs dataset in turbot, we downloaded a total of 24 RNA-seq datasets including 248 samples. Then, the lncRNAs of turbot were identified at the whole genome level for the first time and a total of 12,999 lncRNAs were detected. In addition, DE lncRNAs under each biotic and abiotic stress condition and comprehensive stress-responsive lncRNA candidates under both abiotic and biotic stresses were identified using multiple stress-related RNA-seq datasets. Furthermore, the function of comprehensive stress-responsive lncRNA candidates were predicted by their target genes in lncRNA-mRNA co-expression network. Finally, expression pattern analysis and qPCR validation were conducted to further verify the functions of the comprehensive stress-responsive lncRNA candidates. The results of this study not only provide valuable reference resources for further research on the molecular basis of lncRNAs, but also help to better elucidate the roles of lncRNAs in stress-related regulation of turbot. 

## 2. Results

### 2.1. Genome-Wide Identification of lncRNAs in Turbot

In order to acquire a relatively complete identification and annotation of lncRNAs in the genome of turbot, we collected a total of 24 publicly available turbot RNA-seq datasets including 248 samples (1881.68 Gb data) from the NCBI Sequence Read Archive (SRA) database (Table 1). These datasets represented the largest data collected for the identification of turbot lncRNAs to date.

LncRNAs were identified according to the pipeline shown in Figure 1. First, a total of 119,256 non-redundant transcripts was obtained through transcript assembly and merging. Then, according to the GffCompare [53] comparison results (Figure 2), transcripts with class code of “u, i, j, x, and o” and length greater than 200 bp were consistent with the characteristics of lncRNA transcripts [53] and were screened out. As a result, a total of 95,412 candidate lncRNA transcripts were obtained, of which 8573 were candidate lincRNA transcripts. Finally, through coding potential prediction using the ORF Length and GC content (LGC) [54] and FlExible Extraction of Long non-coding RNAs (FEELnc) [55] software v.0.2.1, a non-redundant lncRNA dataset consisting of 12,999 lncRNAs (29,187 transcripts), including 4107 lincRNAs (6591 transcripts) accounting for 31.59% of all lncRNAs, was obtained for further analysis.

**Table 1 ijms-24-15870-t001:** The details of the RNA-seq datasets used in this study.

Trait		SRA Study	Tissue	Number of Samples	Platform (Illumina)	Size (Gb)	Reference
Crowding	-	SRP129900	kidney, spleen	12	HiSeq 4000	68.2	[56]
Feeding	*myo*-inositol	SRP188583	gill	15	HiSeq 4000	115.45	[57]
	fish meal, soybean meal	SRP074811	intestine	2	NextSeq 500	42.56	[58]
	sodium butyrate, soybean meal	SRP275545	intestine	6	HiSeq 2000	50.23	[59]
Heat	14, 23, 25, 28 °C	SRP152627	kidney	10	HiSeq 4000	88.99	[40]
	14, 20, 24, 28 °C	SRP273870	liver	12	HiSeq 2500	84.49	[60]
Oxygen	-	SRP167318	gill	9	HiSeq 2500	58.99	[43]
Pathogen	*Enteromyxum scophthalmi*	SRP308109	blood	49	HiSeq 4000	381.62	[44]
		SRP255305	thymus	10	HiSeq 4000	17.55	[52]
		SRP065375	kidney, pyloric caeca, spleen	12	HiSeq 2000	31.48	[46]
		SRP050607	kidney, pyloric caeca, spleen	12	HiSeq 2000	36.02	[51]
	*Vibrio anguillarum*	SRP191266	intestine	4	HiSeq 2500	53.34	[45]
		SRP336094	liver	2	NovaSeq 6000	12.68	[47]
		SRP335896	kidney	2	NovaSeq 6000	12.47	[47]
		SRP320422	gill	2	NovaSeq 6000	13.58	[47]
		SRP319434	spleen	2	NovaSeq 6000	12.73	[47]
	Megalocytivirus	SRP347383	kidney	18	HiSeq 2500	172.43	[61]
Salinity	low-salinity	SRP277001	liver	6	HiSeq 4000	49.35	[41]
	low- and high-salinity	SRP238143	gill	9	HiSeq 2000	70.48	[62]
	low- and high-salinity	SRP153594	kidney	9	HiSeq 4000	70.86	[42]
Growth	-	SRP075669	brain, muscle	12	HiSeq 2500	36.61	[63]
Sex	-	SRP136753	testis	18	HiSeq X Ten	120.7	[64]
		SRP261889	testis, ovary	3	HiSeq 4000	83.12	-
		SRP287484	ovary	12	HiSeq 2500	197.75	-
Total		-		248	-	1881.68	

### 2.2. Sequence Characteristics of lncRNA Transcripts in Turbot 

We further analyzed the basic features of lncRNA transcripts and compared them to those of coding gene transcripts. The length of lncRNA transcripts ranged from 200 to 25,059 bp, and the length of lincRNA transcripts ranges from 200 to 10,782 bp (Figure 3A). LncRNA and lincRNA transcripts with sequence length shorter than 1000 bp approximately accounted for 40.82% and 57.91% of the total number of lncRNA and lincRNA transcripts, respectively, which was significantly higher than that of coding genes (9.76%) (Figure 3A). Moreover, the average lengths of lncRNA and lincRNA transcripts were 1627.6 and 1195.8 bp, respectively, which was significantly shorter than that of coding genes (3752.01 bp) (Figure 3B). Furthermore, the GC contents of lncRNA and lincRNA transcripts were 46.65% and 45.34%, respectively, and they were both slightly lower than that of coding genes (50.13%) (Figure 3C). Furthermore, the average expression level (transcripts per million, TPM) of coding genes (TPM = 40.99) was about twice that of lncRNAs (TPM = 20.31), and about 4.4 times that of lincRNA (TPM = 9.23) (Figure 3D).

### 2.3. Tissue-Specific Expression of lncRNAs in Turbot 

The expression patterns of lncRNAs, lincRNAs and coding genes in 12 different tissues, including intestine, liver, ovary, brain, blood, spleen, muscle, testis, gill, pyloric caeca, thymus and kidney, were analyzed using multiple RNA-seq datasets. The results are shown in Figure 4. On the whole, the expression patterns of lncRNAs (Figure 4A) and lincRNAs (Figure 4B) were quite similar, and their overall expression levels were higher in the testis, brain, thymus, gill and kidney tissues, but were lower in the liver and muscle tissues. In comparison, lincRNAs showed stronger tissue-specific expression than lncRNAs. Furthermore, the expression levels of coding genes (Figure 4C) were lower in the liver tissue, but were higher in the testis, brain, thymus, gill, kidney and ovary tissue. The above analysis results showed that lncRNAs, lincRNAs and coding genes all presented tissue-specific expression patterns and were similar to some extent. However, lncRNAs, especially lincRNAs, showed a stronger tissue-specific expression than coding genes.

### 2.4. Differentially Expressed lncRNAs under Biotic Stress Conditions in Turbot

To identify lncRNAs in response to various biotic stresses, ten RNA-seq datasets, including a total of 113 samples related to three different pathogens (*V. anguillarum*, *E. scophthalmi*, and Megalocytivirus) infection stresses, were used to identify DE lncRNAs in multiple tissues under biotic stress conditions. As a result, a total of 2893 DE lncRNAs were detected between each infected group and the control group, among which 612, 1351 and 1060 lncRNAs were differentially expressed under *V. anguillarum*, *E. scophthalmi*, and Megalocytivirus infection stress conditions, respectively. In addition, among these DE lncRNAs, 151 lncRNAs showed differential expression under all three kinds of pathogen infection stress conditions (Figure 5A), indicating that they may have significant responses to various biotic stresses (pathogen infection stresses).

### 2.5. Differentially Expressed lncRNAs under Abiotic Stress Conditions in Turbot

DE lncRNAs in multiple tissues under three different abiotic stress conditions including heat, oxygen, and salinity infection stresses were also identified using six RNA-seq datasets comprising of 55 samples. As a result, a total of 2984 DE lncRNAs was detected between each stress treatment group and the control group, among which 875, 420 and 1689 lncRNAs were differentially expressed under heat, oxygen, and salinity stress condition, respectively. Furthermore, among these DE lncRNAs, 62 lncRNAs showed differential expression under all of the three kinds of abiotic stress conditions (Figure 5B), demonstrating their important roles in response to different abiotic stresses.

In addition, further analysis showed that 11 DE lncRNAs, such as *lnc_MSTRG.17190*, *lnc_MSTRG.28492*, *lnc_MSTRG.22156*, *linc_MSTRG.12463*, *lnc_MSTRG.7047*, *lnc_MSTRG.4117*, *linc_MSTRG.7983*, *lnc_MSTRG.7990*, *linc_MSTRG.13517*, *lnc_MSTRG.9703* and *lnc_MSTRG.17785*, were differentially expressed under all biotic and abiotic stress conditions (Figure 5C), indicating that these 11 lncRNAs may play crucial roles in response to both abiotic and biotic stresses and can be selected as comprehensive stress-responsive lncRNA candidates.

### 2.6. Expression Patterns of Comprehensive Stress-Responsive lncRNA Candidates under Biotic Stress Conditions

To further illustrate the dynamic functions of the above 11 comprehensive stress-responsive lncRNA candidates in response to distinct biotic stresses, we analyzed their expression patterns using *V. anguillarum*, *E. scophthalmi*, and Megalocytivirus infection stress-related RNA-seq datasets.

We first illustrated the expression patterns of these 11 lncRNAs in five different tissues, including the intestine, spleen, gill, kidney and liver tissues, following the *V. anguillarum* challenge (Figure 6A). On the whole, 4, 3, 4, 9 and 8 lncRNAs were differentially expressed in the tissues of the intestine, splenic, gill, kidney and liver, respectively. Specifically, in the intestine, *lnc_MSTRG.17910* and *lnc_MSTRG.7990* had similar expression patterns that continuously significantly upregulated expression with the extension of infection time. Moreover, *linc_MSTRG.12463* was significantly upregulated at 12 h post-infection (hpi) with *V. anguillarum*. *lnc_MSTRG.28492* was continuously downregulated and significantly downregulated at 12 hpi. In the spleen, *lnc_MSTRG.28492* and *lnc_MSTRG.7074* were significantly downregulated; in contrast, *linc_MSTRG.7983* was significantly upregulated. In the gill, *lnc_MSTRG.7047*, *linc_MSTRG.13517* and *lnc_MSTRG.9703* were significantly upregulated, whereas *linc_MSTRG.7983* was significantly downregulated. In the kidney, *lnc_MSTRG.17190*, *lnc_MSTRG.7047*, *lnc_MSTRG.7990*, *linc_MSTRG.13517*, *lnc_MSTRG.9703* and *lnc_MSTRG.17785* showed significantly upregulated expression, but *linc_MSTRG.12463*, *linc_MSTRG.7983* and *lnc_MSTRG.4117* showed the opposite expression patterns. In the liver, *lnc_MSTRG.17190*, *lnc_MSTRG.22156*, *linc_MSTRG.12463*, *lnc_MSTRG.7047*, *linc_MSTRG.13517* and *lnc_MSTRG.9703* were significantly upregulated, while *lnc_MSTRG.28492* and *lnc_MSTRG.7990* were significantly downregulated.

Then, the expression patterns of these 11 lncRNA candidates in the kidney after infection with Megalocytivirus were elucidated (Figure 6B). On the whole, no lncRNA candidate showed differential expression at 3 dpi, whereas 7 and 10 lncRNA candidates showed differential expression at 6 and 9 dpi, respectively. Of them, *lnc_MSTRG.17190*, *lnc_MSTRG.22156*, *linc_MSTRG.13517* and *lnc_MSTRG.9703* were significantly upregulated both at 6 and 9 dpi, whereas *linc_MSTRG.7983* and *lnc_MSTRG.4117* were significantly downregulated both at 6 and 9 dpi. Furthermore, *lnc_MSTRG.17785* was only significantly upregulated at 6 dpi, while *lnc_MSTRG.28492*, *linc_MSTRG.12463*, *lnc_MSTRG.7047* and *lnc_MSTRG.7990* were significantly upregulated at 9 dpi.

In addition, the expression patterns of these 11 lncRNAs in five different tissues, including the kidney, pyloric caeca, spleen, thymus, and blood, after challenging *E. scophthalmi*, were also illustrated (Figure 6C). Overall, 7, 8, 5, 3 and 4 lncRNAs showed differential expression in the kidney, pyloric caeca, spleen, thymus, and blood. In the kidney, only *lnc_MSTRG.17190* showed differential expression at 24 days post-infection (dpi) with *E. scophthalmi*, and the expression level was significantly upregulated. In contrast, five lncRNAs, including *lnc_MSTRG.17190*, *lnc_MSTRG.22156*, *linc_MSTRG.13517*, *lnc_MSTRG.9703* and *lnc_MSTRG.7990*, were significantly upregulated at 42 dpi, while *linc_MSTRG.12463* and *lnc_MSTRG.28492* were significantly downregulated. In the pyloric caeca, no DE lncRNA was found at 24 dpi. In contrast, eight lncRNAs showed differentially expressed expression at 42 dpi, among which *lnc_MSTRG.17190*, *lnc_MSTRG.22156*, *linc_MSTRG.13517* and *lnc_MSTRG.9703* showed significantly upregulated expression, whereas *linc_MSTRG.7983*, *linc_MSTRG.12463*, *lnc_MSTRG.7047* and *lnc_MSTRG.4117* showed significantly downregulated expression. In the spleen, only *lnc_MSTRG.7990* showed differential expression at 24 dpi, and its expression was significantly upregulated. Five lncRNA candidates, including *lnc_MSTRG.17190*, *lnc_MSTRG.7990*, *linc_MSTRG.13517*, *lnc_MSTRG.9703* and *lnc_MSTRG.4117*, showed differential expression at 42 dpi, and they were all significantly upregulated. In the thymus, only three lncRNA candidates, including *linc_MSTRG.13517*, *lnc_MSTRG.4117* and *lnc_MSTRG.17785*, were detected to have differential expression at 42 dpi, and all of them had significantly upregulated expression. In the blood, only four DE lncRNAs, including *lnc_MSTRG.17190*, *lnc_MSTRG.22156*, *linc_MSTRG.13517* and *lnc_MSTRG.9703*, were detected in the *E. scophthalmi* severe infected group, and all of them had significantly upregulated expression.

### 2.7. Expression Patterns of HSP70 Genes under Biotic Stress

To further illustrate the dynamic functions of the above 11 comprehensive stress-responsive lncRNA candidates in response to various abiotic stresses, their expression patterns were illustrated using heat, oxygen, and salinity stress-related RNA-seq datasets.

The expression patterns of 11 lncRNA candidates in the gill under oxygen stress were first clarified (Figure 6D). Specifically, in the air treatment group, *lnc_MSTRG.17190*, *lnc_MSTRG.2849*, *lnc_MSTRG.22156*, *lnc_MSTRG.7047*, *lnc_MSTRG.7990*, *linc_MSTRG.13517* and *lnc_MSTRG.17785* were significantly upregulated, while *linc_MSTRG.12463* and *lnc_MSTRG.4117* were significantly downregulated. By comparison, in the oxygen treatment group, *lnc_MSTRG.17190*, *lnc_MSTRG.28492* and *lnc_MSTRG.9703* had significantly upregulated expression, while *linc_MSTRG.7983* and *lnc_MSTRG.4117* had significantly downregulated expression.

The expression patterns of 11 lncRNA candidates in the kidney and liver tissues under heat treatment stress were also analyzed (Figure 6E). In the kidney, only *lnc_MSTRG.22156* had significantly upregulated expression at 25 °C and 5 lncRNAs including *lnc_MSTRG.17190*, *lnc_MSTRG.22156*, *lnc_MSTRG.7990*, *linc_MSTRG.13517* and *lnc_MSTRG.9703* had significantly upregulated expression at 28 °C. Moreover, *linc_MSTRG.12463*, *linc_MSTRG.7983* and *lnc_MSTRG.4117* had significantly downregulated expressions at 28 °C. In the liver, *lnc_MSTRG.22156* was significantly upregulated at 24 °C and 28 °C, while *lnc_MSTRG.28492* (at 24 °C and 28 °C), *lnc_MSTRG.7047* (at 28 °C) and *lnc_MSTRG.17785* (at 28 °C) were significantly downregulated.

The expression patterns of 11 lncRNA candidates in the kidney, gill and liver tissues under low- and high-salinity stresses were finally elaborated (Figure 6F). On the whole, two, eight and six lncRNA candidates were differentially expressed in the kidney, gill and liver, respectively. In the kidney, no lncRNA candidate was differentially expressed under low-salinity stress, and *lnc_MSTRG.17190* and *lnc_MSTRG.7990* were significantly upregulated under high-salinity stress. In the gill, *lnc_MSTRG.17190*, *lnc_MSTRG.28492*, *linc_MSTRG.13517*, *lnc_MSTRG.9703* and *lnc_MSTRG.4117* were significantly upregulated under low-salinity stress. By contrast, *lnc_MSTRG.7990*, *linc_MSTRG.13517*, *lnc_MSTRG.4117* and *lnc_MSTRG.17785* were significantly upregulated under high-salinity stress, while *lnc_MSTRG.17190* and *linc_MSTRG.12463* showed significantly downregulated expression. In the liver under freshwater treatment, *lnc_MSTRG.17190*, *lnc_MSTRG.28492*, *lnc_MSTRG.22156* and *lnc_MSTRG.7990* were significantly upregulated, and *lnc_MSTRG.7047* and *linc_MSTRG.7983* were significantly upregulated.

### 2.8. Functional Prediction of Comprehensive Stress-Responsive lncRNA Candidates

To obtain further insight regarding the functions of the above 11 comprehensive stress-responsive lncRNA candidates, GO and KEGG enrichment analyses were conducted on their co-expressed target genes. First, according to the correlation coefficients of gene expressions for lncRNA-mRNA pairs, 497 significantly co-expressed target genes (*R* > 0.7 and *q*-value < 0.05) for 11 comprehensive stress-responsive lncRNA candidates were identified, and their interactions were shown in Figure 7. Then, GO enrichment analysis indicated that these target genes were mainly enriched in molecular functions, such as cytokine activity (GO:0005125), glycosaminoglycan binding (GO:0005539), active transmembrane transporter activity (GO:0022804), and in biological processes, such as response to stimulus (GO:0050896), immune response (GO:0006955) (*p* < 0.05) (Figure 8A). Moreover, KEGG enrichment analysis demonstrated that these target genes significantly enriched in pathways, such as protein families: signaling and cellular processes, transporters, metabolism (*p* < 0.05) (Figure 8B). The above results showed that these 11 comprehensive stress-responsive lncRNAs played important roles in response to various stresses in turbot. In addition, some significantly correlated lncRNA-mRNA pairs, such as *lnc_MSTRG.7990*-*DSP6* (dual specificity protein phosphatase 6) (*R* = 0.835, *q*-value = 0), *lnc_MSTRG.7990*-*Cldn4* (claudin-4-like) (*R* = 0.747, *q*-value = 0) and *linc_MSTRG.13517*-*TNIP2* (TNFAIP3-interacting protein 2 isoform X1) (*R* = 0.747, *q*-value = 0), are worth for further investigated, because coding genes in these lncRNA–mRNA pairs have been proven to play important roles in stress response.

### 2.9. qPCR Validation of Comprehensive Stress-Responsive lncRNA Candidates

To further verify the potential comprehensive stress-responsive roles of the above 11 lncRNA candidates in turbot, qPCR validation was performed. First, we validated their expression patterns in the kidney of turbot after heat treatment for 24 h (Figure 9). The qPCR results indicated that the significantly upregulated or downregulated expression patterns of these lncRNAs, except *linc_MSTRG.12463*, were in accordance with the RNA-seq analysis results, which indicate that these comprehensive stress-responsive lncRNA candidates play essential roles in response to abiotic stress, especially heat stress. Then, the expression patterns of these lncRNAs in the kidney of turbot at 0, 6, 12, 24, and 48 h post-infection with *E. tarda* (Figure 10) were also systematically illustrated. They all showed significantly or extremely significantly upregulated expression in the kidney at 6, 12, 24, and 48 h after challenging with *E. tarda*. In addition, except for *lnc_MSTRG.28492* and *linc_MSTRG.12463*, the upregulated or downregulated expression patterns of these lncRNAs were in accordance with the RNA-seq analysis results, demonstrating their crucial roles in response to biotic stress, especially *E. tarda* infection stress.

## 3. Discussion

As a class of non-coding RNA molecules, lncRNAs play crucial roles in a variety of biological processes, such as immune response, growth, development and stress response [26]. In recent years, a great number of studies have demonstrated that lncRNAs indeed exerted crucial roles in response to various abiotic and biotic stresses in multiple teleost species, such as Atlantic salmon [18,38], tilapia [19], large yellow croaker [21], rainbow trout [15], red cusk-eel [17], coho salmon [22]. However, the identification of lncRNAs and their roles in stress response have not been reported in turbot until now. The rapid development of sequencing technologies, the acquisition of high-quality turbot genome sequence, the accumulation of abundant RNA-seq datasets of turbot, and the great improvement of lncRNA identification and functional study methods make it possible for us to carry out systematic identification and functional research of lncRNAs in turbot.

In this study, a lncRNA dataset including 12,999 lncRNAs (containing 29,187 transcripts) was constructed in turbot for the first time using a total of 24 RNA-seq datasets (1.88 TB) consisting of 248 samples from 12 different tissues, which is relatively complete compared with the reported lncRNA datasets in other teleost species. For instance, 21,065 high confidence lncRNAs (transcripts) were identified in Atlantic salmon using RNA-seq datasets from six different tissues of healthy and infectious salmon anemia virus (ISAV) infected individuals [18]. In rainbow trout, 5916 lncRNAs were discovered using RNA-seq datasets of liver tissue under heat stress [15]. Moreover, Leiva et al. identified 4975 lncRNAs using the RNA-seq datasets from the liver, spleen and kidney tissues of silver salmon [22]. Jiang et al. identified a total of 15,147 lncRNAs in large yellow croaker using the RNA-seq datasets from four different tissues including the liver, ovum, spleen and muscle [21]. In addition, Li et al. detected 72,276 high confidence lncRNAs (transcripts) in tilapia using 103 RNA-seq datasets [19]. The above studies showed that more lncRNAs were identified in turbot (this study), large yellow croaker and tilapia, indicating that the more RNA-seq datasets from different experimental conditions and tissue samples were used, the more complete lncRNA datasets can be obtained, which was consistent with the findings in tilapia [19].

The comparison of transcript sequence characteristics between lncRNAs and coding genes showed that the average transcript length of lncRNA (1627.6 bp) was significantly shorter than that of the coding genes (3752.01 bp), and the GC content of the lncRNA transcripts (46.65%) was slightly lower than that of the coding gene transcripts (50.13%), which were in accordance with the results of tilapia [19], Atlantic salmon [18] and red cusk-eel [17]. Meanwhile, a number of studies indicate that mammalian lncRNAs also have similar sequence characteristics [65,66,67]. In addition, in our study, the average expression level of lncRNAs was much lower than that of coding genes, which was in accordance with the results of studies in large yellow croaker [21], Atlantic salmon [38], zebrafish [68], human [69] and mouse [70]. Furthermore, expression pattern analyses of lncRNAs and coding gene in 12 different tissues of turbot showed that lncRNAs, especially lincRNAs, had stronger tissue-specific expression than coding genes, which was consistent with the results of studies on large yellow croak [21], red cusk-eel [17] and zebrafish [68]. In a word, short transcript length, low GC content and tissue-specific expression may be the main characteristics of lncRNAs when compared to coding genes.

In order to demonstrate the important functions of lncRNAs in teleost species in response to various stresses, in recent years an increasing number of studies have detected the DE lncRNAs under stress conditions in multiple teleost species using stress-related RNA-seq datasets, such as juvenile rainbow trout (estradiol exposure stress) [36], rainbow trout (heat stress) [15], zebrafish (spring viraemia of carp virus infection stress) [71], large yellow croaker (*V. parahaemolyticus* infection stress) [37], red common carp (bisphenol A(BPA) exposure stress) [72], Nile tilapia (*Streptococcus agalactiae* infection stress) [73] and red cusk-eel (handling stress) [17]. However, most of the reported studies on identification of stress-responsive lncRNAs in fish were carried out under a single stress condition. Therefore, to explore whether there are lncRNAs that respond to multiple different stresses, in the present study, we identified the DE lncRNAs in response to diverse stress conditions including *V. anguillarum*, *E. scophthalmi* and Megalocytivirus infection, as well as heat, oxygen, and salinity stresses. The results indicated that 151 lncRNAs and 62 lncRNAs were differentially expressed under three different biotic stress conditions and three distinct abiotic stress conditions, respectively. Of those, 11 lncRNAs were differentially expressed under both biotic and abiotic stress conditions. Furthermore, expression pattern analyses showed that most of these 11 lncRNAs were significantly upregulated in different tissues under six stress conditions. To further validate the functions of these 11 lncRNAs, heat stress and *E. tarda* infection stress experiments and qPCR analyses were conducted. The results showed that these 11 lncRNAs were significantly or extremely significantly upregulated or downregulated under the conditions of heat stress and *E. tarda* infection stress. These results indicated that these 11 lncRNAs may play important roles in response to various stresses in turbot and can be selected as comprehensive stress-responsive lncRNA candidates for further targeting and functional verification, thereby elucidating the molecular regulation mechanism of lncRNAs in comprehensive stress response of turbot.

Predicting the putative biological function of lncRNAs is one of the major challenges in the study of lncRNAs. The prediction of the interaction between lncRNAs and mRNAs is of great significance for studying the function of lncRNAs [19]. So, to gain insight into the global functional characterization of these 11 comprehensive stress-responsive lncRNA candidates in this study, we calculated the correlation coefficients of expressions for lncRNA-mRNA pairs, and further performed GO and KEGG enrichment analyses on their significantly correlated target genes. Results indicated that these 11 comprehensive stress-responsive lncRNA candidates are involved in biological process of transmembrane transporter, response to stimulus, signaling and cellular during various stress responses. Furthermore, we further detected a few significantly correlated lncRNA–mRNA pairs that were worthy of further investigation, such as *lnc_MSTRG.7990*-*DSP6*, *lnc_MSTRG.7990*-*Cldn4*, *linc_MSTRG.13517*-*TNIP2*. Previous studies have demonstrated that target genes (*DSP6*, *Cldn4*, *TNIP2*) in these lncRNA–mRNA pairs were involved in stress response [74,75,76]. For instance, *TNIP2* can negatively and positively regulate transcription of the NF-κB-dependent target genes, and the transcription factors NF-κB family play a crucial role in regulating cellular responses to environmental stresses [74]. Moreover, the expression of DSP genes is strongly induced by a variety of cellular stresses or growth factors, and DSP genes can control the function of MAP kinases, which play important roles in a series of stress-activated signal transduction pathways [75]. In addition, *Cldn4* was identified as an osmotic reactive protein in inner medullary collecting duct 3 (IMCD3) cells and the papilla of mouse kidneys, and its expression level significantly increased under hypertonic stress [76]. The above results demonstrated preliminarily the potential regulatory roles of lncRNAs on coding genes in response to various stresses. However, to comprehensively explore the regulatory function of these lncRNAs on mRNAs, the further targeted validations and research on the above and other significantly correlated lncRNA–mRNA interaction pairs are needed in the future.

## 4. Materials and Methods

### 4.1. RNA-seq Datasets Used for the Identification of lncRNAs in Turbot

In order to conduct the most comprehensive identification and annotation of lncRNAs in the turbot genome, we downloaded all available published turbot RNA-seq datasets from the NCBI SRA database with SRAtoolkit software (v2.11.0) [77]. A total of 24 RNA-seq datasets associated with 8 traits, including sex, growth, crowding, feeding, heat stress, oxygen stress, salinity stress and pathogens (*E. scophthalmi*, *V.* anguillarum, and Megalocytivirus) infection stress were acquired. The details of the RNA-seq datasets used in the present study are shown in Table 1.

### 4.2. Bioinformatics Pipeline for Identifying lncRNAs

All above 24 RNA-seq datasets were used to identify the lncRNAs in turbot. The pipeline of lncRNA identification is shown in Figure 1. First, fastq-dump tool of SRAtoolkit software was used to convert the RNA-seq datasets downloaded in SRA format into FASTQ format [77]. Then, reads were aligned with the latest assembled turbot genome (GCA_022379125.1 [50]) using STAR software version 2.7.11a [78] with default parameters. To find new splice junctions, a two-pass alignment for each read was run using STAR with the parameter “--sjdbFileChrStartEnd”. The aligned reads were assembled into transcripts using StringTie [79] with default parameters, and all assembled transcripts were merged into a final non-redundant transcriptome using StringTie with “--merge” mode. Then, the merged transcripts were compared to the genome gene transfer format (GTF) file using GffCompare [53] to determine their genomic locations with respect to the known encoding gene transcripts on the genome. According to the results of GffCompare comparison, transcripts with the class code of “u, i, j, x, and o” and length greater than 200 bp were selected as candidate lncRNA transcripts. Of these, transcripts with class code of “u” were candidate transcripts of intergenic long non-coding RNA (lincRNA). Finally, the coding potential of candidate transcripts was predicted using LGC [54] and FEELnc [55] software version 2.0.6, and only transcripts with non-coding potential were retained for further analysis.

### 4.3. Analysis of Sequence Characteristics of lncRNA Transcripts

The sequence characteristics of lncRNA and lincRNA transcripts were analyzed and compared with those of coding genes in terms of average sequence length, GC content and expression level (TPM).

### 4.4. Tissue Expression Analysis of lncRNAs

RNA-seq data from 12 different tissues, including ovary, liver, pyloric caeca, blood, intestine, muscle, testis, thymus, spleen, brain, kidney, and gill of healthy turbot samples under normal conditions (in the control groups) (Table 1) were used to clarify the expression patterns of lncRNAs in different tissues of turbot. First, the TPM values of lncRNAs, lincRNAs and coding genes in different tissues were calculated by TPMCalculator [80] using the sorted BAM file obtained from STAR alignment in 4.2. Then, lncRNAs, lincRNAs and coding genes that were expressed in at least one tissue (that is, the expression level was greater than 0 in at least one tissue) were selected, and their TPM values were normalized into log_2_(TPM + 1) to draw lncRNA expression heat maps.

### 4.5. Identification of Differentially Expressed lncRNAs under Abiotic and Biotic Stresses

In this study, DE lncRNAs under biotic (pathogen infection) and abiotic stresses were identified using RNA-seq datasets related to three different pathogens (*E. scophthalmi*, *V.* anguillarum, and Megalocytivirus) infection stresses and three distinct abiotic stresses (heat, oxygen, and salinity), respectively. First, we constructed read count matrixes using featureCounts [81] software program version 2.0.6 in the Subread [82] package with the sorted BAM files acquired from STAR alignment in Section 4.2. edgeR [83] was then used to identify the DE lncRNAs (false discovery rate (FDR) < 0.05 and |log_2_FC| > 1) between the control group and the stress treatment group under all biotic and abiotic stress conditions. Finally, lncRNAs that were differentially expressed under both biotic and abiotic stresses were selected as comprehensive stress-responsive lncRNA candidates.

### 4.6. Expression Patterns of the Comprehensive Stress-Responsive lncRNA Candidates

In order to better elucidate the roles of comprehensive stress-responsive lncRNA candidates in response to distinct stresses, their expression patterns under six stress conditions including *E. scophthalmi*, *V. anguillarum*, and Megalocytivirus infection stress, and heat, oxygen, and salinity stress were illustrated using heat maps with normalized TPM values (log_2_(TPM + 1)) of lncRNAs. The detailed methods were included in Section 4.4.

### 4.7. Functional Prediction of the Comprehensive Stress-Responsive lncRNA Candidates

We further predict the putative biological functions of comprehensive stress-responsive lncRNA candidates based on the functional annotations of their co-expressed target coding genes. First, the R program Hmisc (https://hbiostat.org/R/Hmisc/) accessed on 10 April 2023 was used to calculate the correlation coefficients (*R*) of expression for comprehensive stress-responsive lncRNAs and mRNAs pairwise. Then, the significance of differences in correlation coefficients were evaluated using Benjamini-Hochberg (BH) method. The target genes in significantly correlated comprehensive stress-responsive lncRNA-mRNA pairs (*R* > 0.7 and *q*-value < 0.05) were selected for GO and KEGG enrichment analyses. Moreover, the significantly correlated comprehensive stress-responsive lncRNA-mRNA pairs were displayed using Cytoscape 3.9.1 [84].

### 4.8. Expression Analyses of HSP70 Genes

Samples acquired from *E. tarda* infection stress and heat stress experiments in turbot in our previous study [49] were used to validate the expressions of comprehensive stress-responsive lncRNA candidates. The turbot used in this study were purchased from Haiyang Yellow Sea Aquatic Product Co., Ltd, Yantai, China. All fish samples collection and handling in the present study conformed to the ethical principles of the Animal Care and Use Committee of Yellow Sea Fisheries Research Institute, Chinese Academy of Fishery Sciences (CAFS) (YSFRI-2023004). All experimental methods were approved by the Animal Care and Use Committee of Yellow Sea Fisheries Research Institute, CAFS. The details of the experiments are described as follows.

In the *E. tarda* infection experiment, 60 healthy turbot individuals with mean weight of 25 ± 2.45 g were selected and randomly divided into control group and *E. tarda* challenge group, and they were maintained at 18 °C for 7 days in 500 L aerated water tank. In the *E. tarda* infection group, the fish were injected with 100 µL of *E. tarda* suspension (10^7^ CFU/mL) per 1 g body weight. In the control group, the fish were injected with equal amounts of 1x PBS solution. Three turbot individuals were randomly sampled from each group at 0, 6, 12, 24, and 48 h after injection with *E. tarda*. Fish were anesthetized with clove oil for sample collection. The kidney tissues were then sampled and rapidly frozen in liquid nitrogen, then transferred to −80 °C for storage until RNA extraction.

For the heat stress experiment, 80 turbot individuals with an average weight of 26 ± 2.02 g were selected and randomly divided into four groups. After acclimating at 18 °C for 7 days, the water temperature was raised at a constant rate of 1 °C/h until the predetermined temperature (22 °C (T1), 26 °C (T2), and 30 °C (T3)) were reached, while the control group (C) remained at 18 °C. After heat stress treatment for 24 h, three turbot individuals from each group were randomly selected to collect the kidney tissues. The detail method for sampling was the same as that conducted in the *E. tarda* challenge experiment.

### 4.9. qPCR Validation of the Comprehensive Stress-Responsive lncRNA Candidates 

To further verify the potential roles of the comprehensive stress-responsive lncRNA candidates in response to abiotic and biotic stresses in turbot, qPCR validation was conducted. First, Primer-BLAST from NCBI were used to design the primers of lncRNAs and β-actin (the internal control), and the sequences of primer pairs are shown in Table 2. qPCR experiments were conducted on Applied Biosystems™ 7500 Fast Real-Time PCR System (ABI, Los Angeles, CA, USA) with the THUNDERBIRD^®^ Next SYBR^®^ qPCR Mix (TOYOBO, Osaka, Japan). The relative expression levels of lncRNAs were calculated with the 2^−ΔΔCt^ method. Then, SPSS 26.0 software was used for statistical analysis of the relative expression of lncRNAs by One-Way ANOVA, and *p* value < 0.05 indicates significant difference (*) and *p* value < 0.01 indicates extremely significant difference (**).

## 5. Conclusions

In the present study, we integrated all available RNA-seq datasets of turbot to identify lncRNAs. As a result, a total of 12,999 lncRNAs, including 29,187 transcripts, were uncovered in the turbot reference genome. These lncRNA transcripts showed a shorter average length, lower average GC content and expression levels, as well as stronger tissue-specific expression than coding genes. Moreover, DE analysis identified 151 and 62 DE lncRNAs under various abiotic and biotic stresses, respectively, among which 11 lncRNAs were differently expressed under both abiotic and biotic stresses. Furthermore, the expression patterns, qPCR, co-expression and functional enrichment analyses were all indicated the potential comprehensive stress-responsive functions of these 11 lncRNAs. We present the first comprehensive annotation of lncRNAs in turbot, which not only provide valuable reference resources for further functional research of lncRNAs in turbot, but also lay an important foundation for the development of molecular selective breeding of stress-resistant turbot strains or varieties.

## Figures and Tables

**Figure 1 ijms-24-15870-f001:**
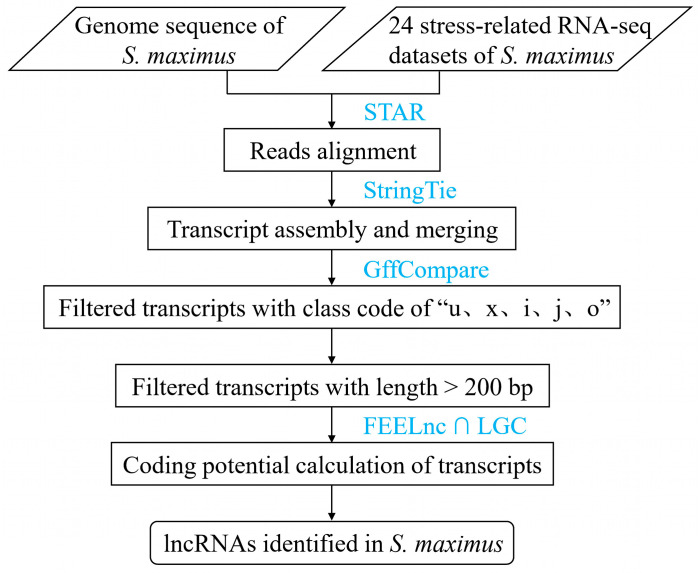
The pipeline of lncRNAs identification in *S. maximus*.

**Figure 2 ijms-24-15870-f002:**
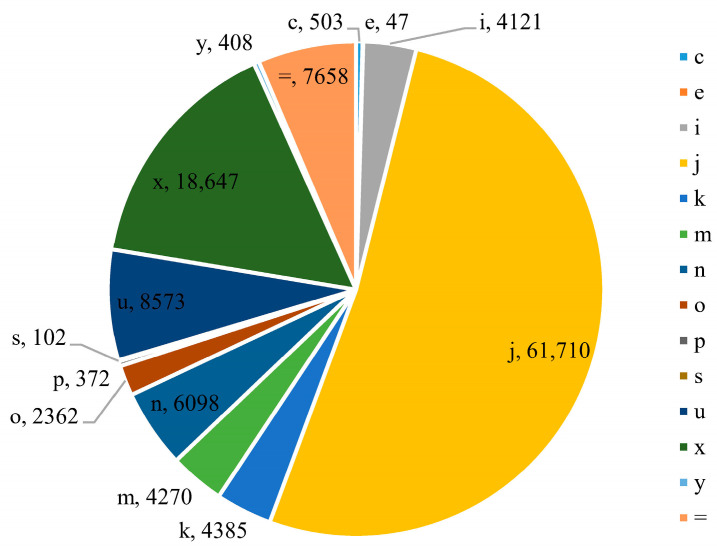
Classification of all assembled transcripts base on GffCompare comparison results. The class codes correspond as follows: “c”, contained in reference (intron compatible); “e”, single exon transfrag partially covering an intron, possible pre-mRNA fragment; “i”, fully contained within a reference intron; “j”, multi-exon with at least one junction match; “k”, containment of reference (reverse containment); “m”, retained intron (s), all introns matched or retained; “n”, retained introns (s), not all introns matched/covered; “o”, other same strand overlap with reference exons; “p”, possible polymerase run-on (no actual overlap); “s”, intron match on the opposite strand (likely a mapping error); “u”, unknown, intergenic transcript; “x”, exonic overlap on the opposite strand (like “o” or “e” but on the opposite strand); “y”, contains a reference within its intron; “=”, complete, exact match with intron chain.

**Figure 3 ijms-24-15870-f003:**
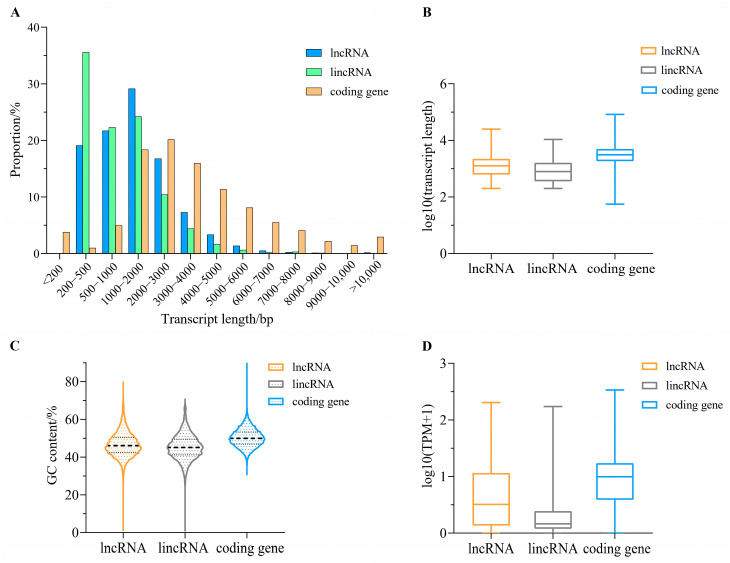
The comparison of lncRNAs, lincRNAs and encoding genes in *S. maximus*. (**A**) The comparison of transcript length distribution, (**B**) The comparison of transcript length, (**C**) The comparison of GC content, (**D**) The comparison of expression level (TPM).

**Figure 4 ijms-24-15870-f004:**
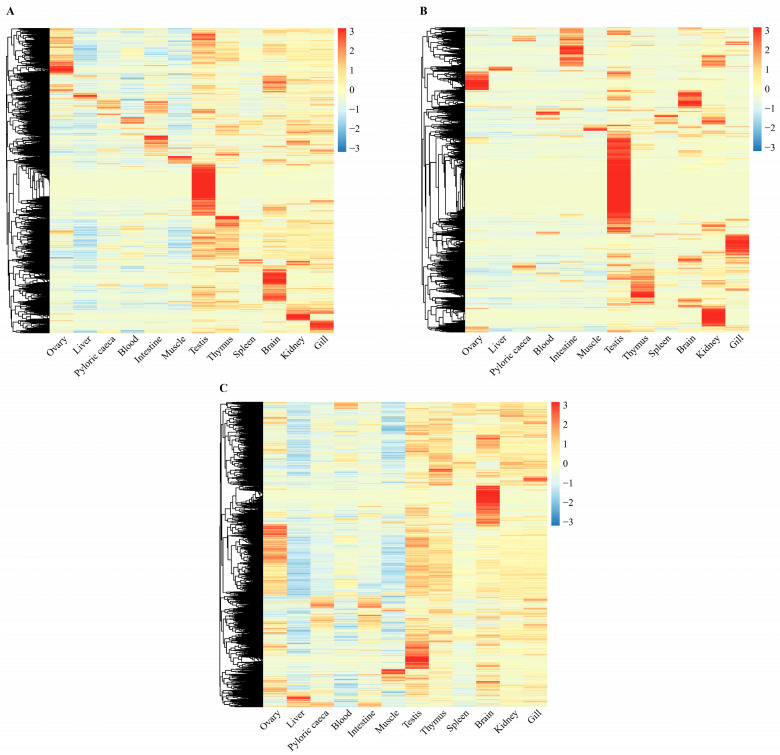
Expression patterns of lncRNAs, lincRNAs and coding genes of turbot in twelve different tissues in *S. maximus*. Cells with different color in the heat maps correspond to different expression values, which were normalized into log_2_ (TPM + 1). Normalized expression values were clustered by “complete” method. (**A**–**C**) represent the expression patterns of lncRNAs, lincRNAs, and coding genes, respectively.

**Figure 5 ijms-24-15870-f005:**
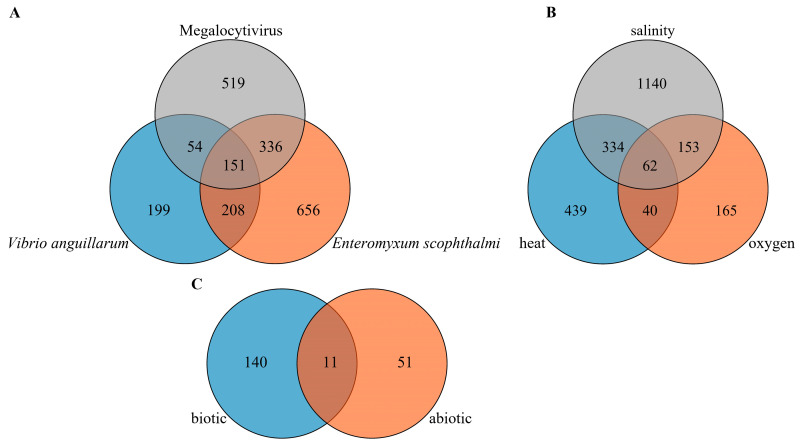
Differentially expressed lncRNAs of turbot under various abiotic and abiotic stresses in *S. maximus*. (**A**) DE lncRNAs under three biotic stresses (*V. anguillarum*, *E. scophthalmi*, and Megalocytivirus infection stress), (**B**) DE lncRNAs under three abiotic stresses (heat, oxygen and salinity stress). (**C**) DE lncRNAs under all biotic and abiotic stress conditions. LncRNAs with false discovery rate (FDR) < 0.05 and |log_2_fold change (FC)| > 1 were defined as DE lncRNAs.

**Figure 6 ijms-24-15870-f006:**
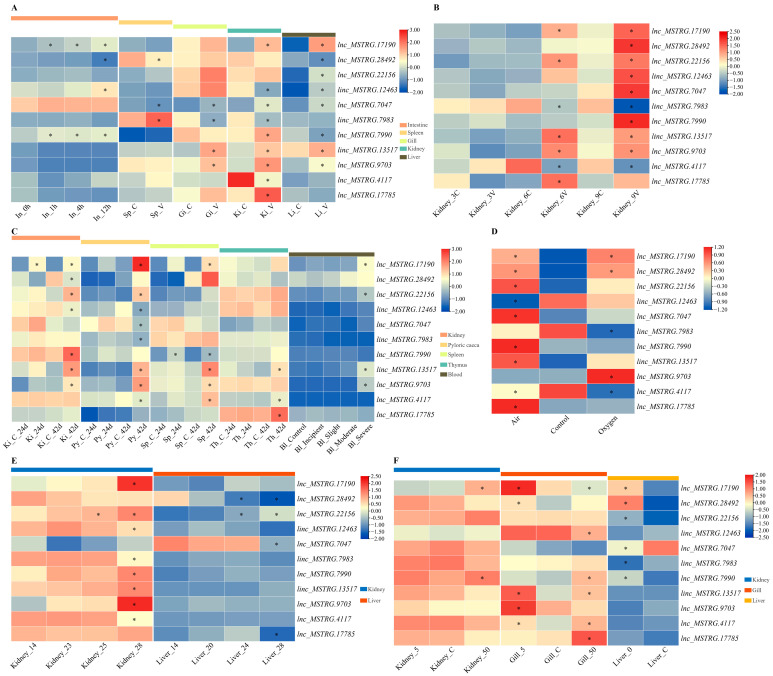
Expression patterns of the 11 comprehensive stress-responsive lncRNA candidates under various abiotic and biotic stresses in *S. maximus*. (**A**) Expression patterns under *V. anguillarum* infection stress. In, Sp, Gi, Ki and Li represented intestine, spleen, gill, kidney and liver, respectively. (**B**) Expression patterns under Megalocytivirus infection stress. C and V represented the control and Megalocytivirus infection group, respectively. (**C**) Expression patterns under *E. scophthalmi* infection stress. Ki, Py, Sp, Th, and Bl represented kidney, pyloric caeca, spleen, thymus, and blood, respectively. (**D**) Expression patterns under oxygen stress. (**E**) Expression patterns under heat stress. (**F**) Expression patterns under salinity stress. * indicated the DE lncRNAs with |log_2_fold change (FC)| > 1 and false discovery rate (FDR) < 0.05.

**Figure 7 ijms-24-15870-f007:**
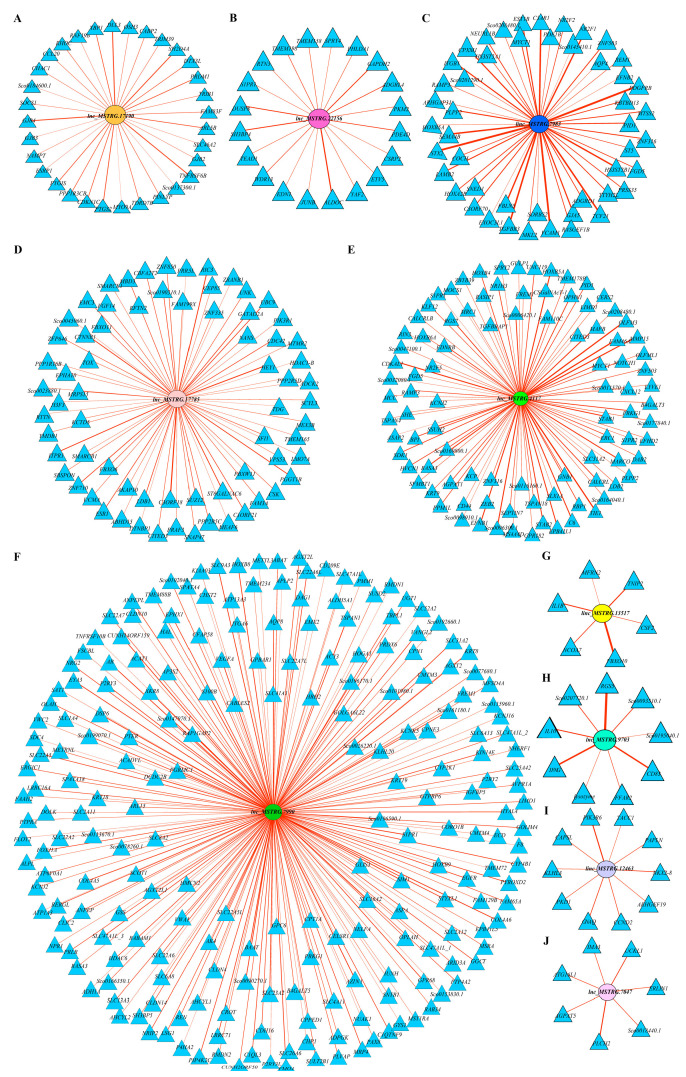
The co-expression network between 10 comprehensive stress-responsive lncRNAs and coding genes in *S. maximus*. (**A**–**J**) represent *lnc_MSTRG.17190*, *lnc_MSTRG.22156*, *linc_MSTRG.7983*, *lnc_MSTRG.17785*, *lnc_MSTRG.4117*, *lnc_MSTRG.7990*, *linc_MSTRG.13517*, *lnc_MSTRG.9703*, *linc_MSTRG.12463*, and *lnc_MSTRG.7047*, respectively. No coding gene is significantly related to *linc_MSTRG.28492*. Elliptical and triangular nodes represent lncRNAs and coding genes, respectively. The edges between nodes represent the interaction relationships of lncRNAs and coding genes, and the width of edges indicate the correlation strength between lncRNAs and coding genes. The co-expression networks were all shown using the layout of “Edge-weighted Spring-Embedded Layout”.

**Figure 8 ijms-24-15870-f008:**
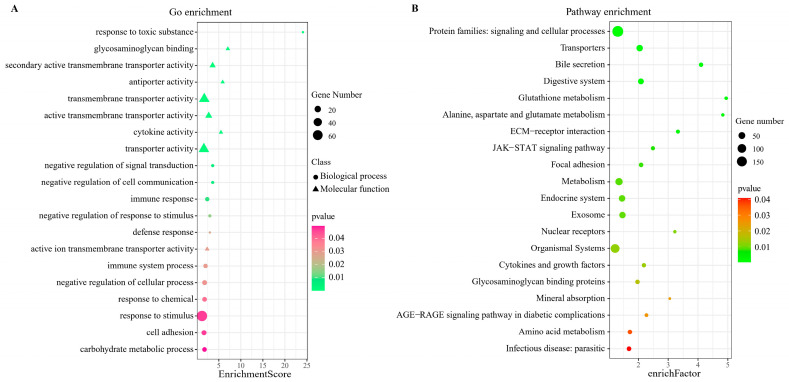
GO and KEGG enrichment analyses of target genes of 11 comprehensive stress responsive lncRNA candidates. (**A**) GO enrichment analysis. (**B**) KEGG enrichment analysis.

**Figure 9 ijms-24-15870-f009:**
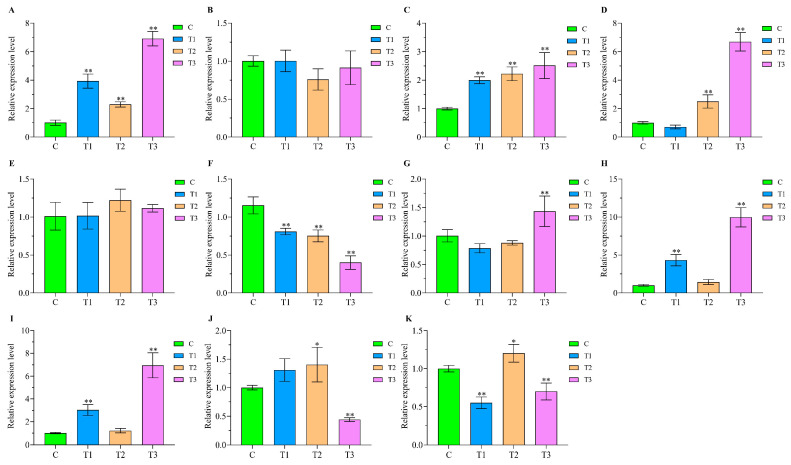
qPCR validation of 11 comprehensive stress-responsive lncRNA candidates in the kidney at 24 h after heat stress. C, T1, T2, and T3 represent the 18 °C, 22 °C, 26 °C, and 30 °C heat groups, respectively. (**A**–**K**) represent *lnc_MSTRG.17190*, *lnc_MSTRG.28492*, *lnc_MSTRG.22156*, *linc_MSTRG.12463*, *lnc_MSTRG.7047*, *lnc_MSTRG.4117*, *linc_MSTRG.7983*, *lnc_MSTRG.7990*, *linc_MSTRG.13517*, *lnc_MSTRG.9703*, and *lnc_MSTRG.17785*, respectively. The relative expression levels of lncRNAs were measured by qPCR experiment with the 2^−ΔΔCt^ method (the same below). * and ** indicate the significant difference (*p* < 0.05) and extremely significant difference (*p* < 0.01) between the control (C) and each heat groups (T1, T2, and T3), respectively.

**Figure 10 ijms-24-15870-f010:**
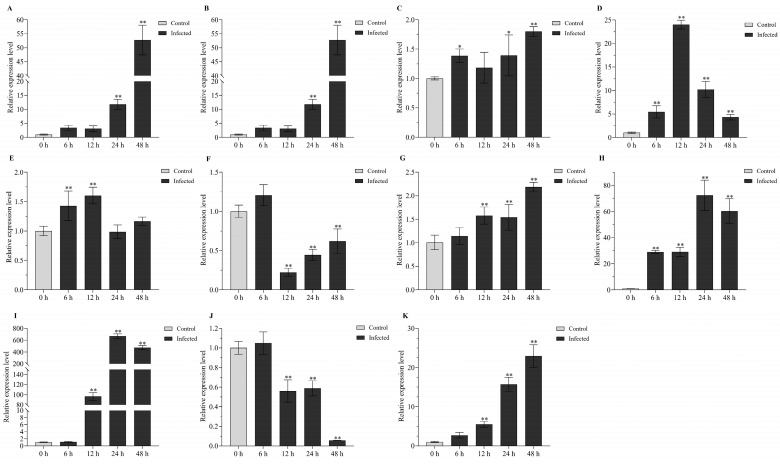
qPCR validation of 11 comprehensive stress-responsive lncRNA candidates in the kidney at 0, 6, 12, 24, and 48 h post infection with *E. tarda*. (**A**–**K**) represent *lnc_MSTRG.17190*, *lnc_MSTRG.28492*, *lnc_MSTRG.22156*, *linc_MSTRG.12463*, *lnc_MSTRG.7047*, *lnc_MSTRG.4117*, *linc_MSTRG.7983*, *lnc_MSTRG.7990*, *linc_MSTRG.13517*, *lnc_MSTRG.9703*, and *lnc_MSTRG.17785*, respectively. * and ** indicate the significant difference (*p* < 0.05) and extremely significant difference (*p* < 0.01) between the control and each infected groups, respectively.

**Table 2 ijms-24-15870-t002:** The primer sequences used to amplify the comprehensive stress-responsive lncRNAs and *β-actin*.

lncRNA or Gene	Primers	Sequences (5′–3′)	The Length of Product/bp
*lnc_MSTRG.17190*	17190-F	CGAACGCTCACAGGAGACTG	20
	17190-R	ACAACTCCACAACCTCACCTG	21
*lnc_MSTRG.28492*	28492-F	CCATACCCGCGATCTGAAGG	20
	28492-R	ATCTTCGAGAGCGTCAACCA	20
*lnc_MSTRG.22156*	22156-F	GCGCACTTTCTTGACACAGG	20
	22156-R	CGGCTGGTGCCTAACTAGAG	20
*linc_MSTRG.12463*	12463-F	GCAAACCTGAAGGAGTAGGCT	21
	12463-R	CTAGATAGGCAGGCCTTGGTC	21
*lnc_MSTRG.7047*	7047-F	ATAAGTAGCCAGCCGTCGAG	20
	7047-R	TGGTGCTAGGTTGAATGCTGT	21
*linc_MSTRG.7983*	7983-F	TCATTCGATTTCACGCACGC	20
	7983-R	GCCTCAAGAAGCTGAGAGCA	20
*lnc_MSTRG.7990*	7990-F	GCTGTAAAGAGCGCTGCAAG	20
	7990-R	AGCTGGTGTCTGAACGACAG	20
*linc_MSTRG.13517*	13517-F	GCGTAGTTGACGTTGGACTT	20
	13517-R	GAGGATCACTGCGGCTACG	19
*lnc_MSTRG.9703*	9703-F	CACGTCGGCACAATCACAA	19
	9703-R	ACGACTTTATGAACAGTGGCA	21
*lnc_MSTRG.4117*	4117-F	TTCTTCTGGTCCTCCTTGCG	20
	4117-R	TCAGCCTCGTGACCTTGAAC	20
*lnc_MSTRG.17785*	17785-F	CGTCTCCTCTCACTGCTCCA	20
	17785-R	CTGAGCTCCTCCACCACGTC	20
*β-actin*	β-F	ACAACGGATCCGGTATGTGC	20
	β-R	CTCTGGGCTTCATCACCTACG	21

## Data Availability

The datasets analyzed for this study can be found in NCBI: https://www.ncbi.nlm.nih.gov/ (accessed on 7 March 2022). The accession numbers can be found below: SRP129900, SRP188583, SRP074811, SRP275545, SRP152627, SRP273870, SRP167318, SRP308109, SRP255305, SRP065375, SRP050607, SRP191266, SRP336094, SRP335896, SRP320422, SRP319434, SRP347383, SRP277001, SRP238143, SRP153594, SRP075669, SRP136753, SRP261889, SRP287484.

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
