# Peer review of "Genome-Wide Identification and Involvement in Response to Biotic and Abiotic Stresses of lncRNAs in Turbot (Scophthalmus maximus)"

_ijms, 2023, doi:10.3390/ijms242115870_

Round 1

Reviewer 1 Report

Comments and Suggestions for Authors

In this manuscript, the author examined the expression patterns of long noncoding RNAs (lncRNAs) in 12 different tissues derived from turbot under abiotic and biotic stress conditions and found lncRNAs that are differentially expressed in a stress type-specific manner.

Line120:

What is “GffCompare” ?

In this “GffoCompare”, an explanation of why only the five class codes u, I, j, x, and o were extracted should also be provided. The explanation of each class code should also be described in this section.

In the “Results” section, lncRNA and lincRNA are compared. Do the lncRNAs used in these analyzes also include lincRNAs, or are they using lncRNAs without lincRNAs?

Are the intronic lncRNAs, antisense lncRNAs, and overlapping lncRNAs described in lines 47 and 48 included in the lncRNA data used in this analysis? In that case, comparisons should also be made for each LncRNA in all analyses.

In Fig. 5

Figures 5A and 5B are reversed.

In Lines 212 and 233, Fig. 5A is described as the result of biotic stress (pathogen infection stress), and in Lines 222 and 234, Fig. 5B is described as the result of abiotic stress. However, Fig. 5 depicts “A“ as a result of abiotic stress and “B” as a result of biotic stress.

Author Response

1. Line120:

What is “GffCompare”?

In this “GffoCompare”, an explanation of why only the five class codes u, i, j, x, and o were extracted should also be provided. The explanation of each class code should also be described in this section.

Response:

Thanks for your suggestion.

“GffCompare” is a generic, standalone tool for merging and tracking transcript structures across multiple samples and comparing them to a reference annotation. GffCompare has the following main functions:

1) merge structurally equivalent transcripts and transcript fragments (transfrags) across multiple samples;

2) assess the accuracy of the assembled transcripts from an RNA-Seq sample by comparing it to known annotation; and

3) track, annotate, and report all structurally distinct transfrags across multiple samples.

The last two purposes require the user to provide a known reference annotation file that GffCompare then uses to classify all the transcripts in the input samples according to the reference transcript that they most closely overlap.

The explanation of each class code have been described in Figure 2, so, we do not describe it in detail in the main body.

The class codes of “u, i, j, x, and o” correspond as follows: “i”, fully contained within a reference intron; “j”, multi-exon with at least one junction match; “u”, unknown, intergenic transcript; “x”, exonic overlap on the opposite strand (like “o” or “e” but on the opposite strand); “o”, other same strand overlap with reference exons. According to the relevant research literatures, transcripts with the class codes of “u, i, j, x, and o” are consistent with the characteristics of lncRNA transcripts, so they were extracted as candidate lncRNA transcript. We have made an explanation in line 121-122.

2. In the “Results” section, lncRNA and lincRNA are compared. Do the lncRNAs used in these analyzes also include lincRNAs, or are they using lncRNAs without lincRNAs?

Are the intronic lncRNAs, antisense lncRNAs, and overlapping lncRNAs described in lines 47 and 48 included in the lncRNA data used in this analysis? In that case, comparisons should also be made for each LncRNA in all analyses.

Response:

The lncRNAs used in these analyzes include lincRNAs.

The lncRNA data used in this analysis included the intronic lncRNAs, antisense lncRNAs, and overlapping lncRNAs. However, long intergenic noncoding RNAs (lincRNAs) are long noncoding transcripts (>200 nt) from the intergenic regions of annotated protein-coding genes. Among all kinds of lncRNAs, lincRNAs have the most clear relationship with the location of coding genes, which is very conducive to the study of their regulatory effects on gene expression. Furthermore, lincRNAs are the focus of our follow-up research, so we took lincRNA separately for comparison, and we have no research plan on intronic lncRNAs, antisense lncRNAs, overlap-ping lncRNAs at present.

3. In Fig. 5

Figures 5A and 5B are reversed.

In Lines 212 and 233, Fig. 5A is described as the result of biotic stress (pathogen infection stress), and in Lines 222 and 234, Fig. 5B is described as the result of abiotic stress. However, Fig. 5 depicts “A“ as a result of abiotic stress and “B” as a result of biotic stress.

Response:

Thanks for your suggestion. We have replaced and resubmitted the Figure 5 in the manuscript.

Reviewer 2 Report

Comments and Suggestions for Authors

The manuscript titled: „Genome-wide Identification and Involvement in Response to 2 Biotic and Abiotic Stresses of lncRNAs in Turbot (Scophthal-3 mus Maximus)”presents bioinformatical research with some experimental/laboratory work. Unfortunately, I cannot accept this publication for several reasons:

1. The lack of experimental design in work where the impact of biotic and abiotic stresses on living organisms was examined is unacceptable.

2. Lack of research hypotheses.

3. Lack of detailed information about experimental animals, experimental groups, their origin, and information about the consent of the ethics committee to conduct research on these animals.

4. The research methodology is imprecise, for example in the case of experiments with the application of different temperatures, other water parameters should also be analyzed because they change together with temperature changes.

5. Without establishing a specific experimental design, only observations are described that can be included in local journals or included as part of a review paper.

6. The results presented in this paper are not helpful to other researchers because because the research contains many methodological errors.

Author Response

The manuscript titled: “Genome-wide Identification and Involvement in Response to Biotic and Abiotic Stresses of lncRNAs in Turbot (Scophthalmus Maximus)” presents bioinformatical research with some experimental/laboratory work. Unfortunately, I cannot accept this publication for several reasons:

1. The lack of experimental design in work where the impact of biotic and abiotic stresses on living organisms was examined is unacceptable.

Response:

Thanks for your suggestion.

We first used bioinformatics methods to identify, characterize and analyze differential expression of lncRNAs in turbot using multiple RNA-seq dataset derived from NCBI SRA database. As a result, 11 potential comprehensive stress responsive lncRNA candidates differentially expressed under both abiotic and biotic stresses were detected. Then, to validate their comprehensive stress responsive functions, we design Edwardsiella tarda infection stress (biotic stress) and heat stress (abiotic stress) experiment and qPCR experiment. Of course, the design of experiments may need to be improved.

2. Lack of research hypotheses.

Response:

We identified 11 differentially expressed lncRNAs under both abiotic and biotic stresses using multiple stress-related RNA-seq datasets. Then, we hypothesize these 11 lncRNAs may be potential comprehensive stress responsive lncRNA candidates, and validate their functions using heat and E. tarda infection stress experiment and qPCR experiment.

3. Lack of detailed information about experimental animals, experimental groups, their origin, and information about the consent of the ethics committee to conduct research on these animals.

Response:

Thanks for your suggestion.

Samples used to validate the expressions of comprehensive stress responsive lncRNA candidates were acquired from E. tarda infection stress and heat stress experiments in our previous study, so we cited the reference (reference 49) and didn’t describe the experimental animals, experimental groups, their origin in detail.

1) experimental animals and experimental groups

In the E. tarda infection experiment, 60 healthy turbot individuals with mean weight of 25 ± 2.45 g were selected and randomly divided into control group and E. tarda challenge group, and they were maintained at 18 â—¦C for 7 days in 500 L aerated water tank. (line 572-574)

For heat stress experiment, 80 turbot individuals with an average weight of 26 ± 2.02 g were selected and randomly divided into four groups. (line 581-582)

2) the origin of experimental animals

The experimental turbot was purchased from Haiyang Yellow Sea Aquatic Product Co., Ltd. We have added this information in the manuscript in line 565-566.

3) the consent of the ethics committee to conduct research on these animals

All fish samples collection and handling in the present study conformed to the ethical principles of the Animal Care and Use Committee of Yellow Sea Fisheries Research Institute, Chinese Academy of Fishery Sciences (CAFS) (YSFRI-2023004). All experimental methods were approved by the Animal Care and Use Committee of Yellow Sea Fisheries Research Institute, CAFS. We have provided this information in the manuscript in line 566-570.

4. The research methodology is imprecise, for example in the case of experiments with the application of different temperatures, other water parameters should also be analyzed because they change together with temperature changes.

Response:

In the heat treatment experiment in this study, we have done our best to ensure other environmental conditions to remain consistent. As you said, other water parameters may be changed because of the changes of temperature. But, the main variable factor in this experiment is temperature. Therefore, the influence of other water parameters on turbot is not considered in this study.

5. Without establishing a specific experimental design, only observations are described that can be included in local journals or included as part of a review paper.

Response:

Thanks for your suggestions.

In this study, we comprehensively analyzed multiple biotic and abiotic stress-related RNA-seq dataset derived from previous studies of turbot, and identified, characterized and analyzed differential expression of lncRNAs for the first time in turbot. As a result, 11 potential comprehensive stress responsive lncRNA candidates differentially expressed under both abiotic and biotic stresses were identified, which may be used as important target lncRNAs in the molecular selective breeding of stress-resistant turbot strains or varieties in the future. So, we design E. tarda infection stress (biotic stress) and heat stress (abiotic stress) experiment and qPCR experiment to validate their comprehensive stress responsive functions preliminary. Certainly, in our follow-up study plan, we will conduct in-depth functional verification and study on each comprehensive stress responsive lncRNA candidates.

6. The results presented in this paper are not helpful to other researchers because the research contains many methodological errors.

Response:

The experimental methods in this paper are all supported by references. Of course, not all methods are necessarily correct. We will further improve and optimize them in the follow-up study according to your suggestions. Thanks a lot.

Round 2

Reviewer 1 Report

Comments and Suggestions for Authors

Their response adequately addresses my question.

Reviewer 2 Report

Comments and Suggestions for Authors

The authors clarified the missing information. However, for the future, I suggest planning the experiment very well, especially when examining the impact of abiotic factors at various levels.